Repeatability of an attention bias test for sheep suggests variable influence of state and trait affect on behaviour

http://orcid.org/0000-0002-4571-2285 Monk Jessica E. 1 2 jessica.monk@csiro.au
http://orcid.org/0000-0001-9497-5148 Colditz Ian G. 2
Clark Sam 1
Lee Caroline 1 2
1 School of Environmental and Rural Science, University of New England , Armidale, NSW , Australia
2 Agriculture and Food, Commonwealth Scientific and Industrial Research Organisation , Armidale, NSW , Australia
Vonk Jennifer
Electronic publication date: 2023 Feb 2
Publication date: 2023
Volume: 11
Electronic Location ID: e14730
Received 2022 Oct 18; Accepted 2022 Dec 20
Copyright: © 2023 Monk et al.
Copyright year: 2023
Copyright holder: Monk et al.
License: This is an open access article distributed under the terms of the Creative Commons Attribution License, which permits unrestricted use, distribution, reproduction and adaptation in any medium and for any purpose provided that it is properly attributed. For attribution, the original author(s), title, publication source (PeerJ) and either DOI or URL of the article must be cited.
License URL: https://creativecommons.org/licenses/by/4.0/

Keywords: Cognitive bias, Vigilance, Animal welfare, Threat perception, Personality, Temperament, Behavior, Livestock, Anxiety, Habituation

Funding: Commonwealth Scientific and Industrial Research Organisation (CSIRO) University of New England School of Environmental and Rural Science Sheep Cooperative Research Centre Top-Up scholarship. Commonwealth Government Australian Postgraduate Award This work was supported by the Commonwealth Scientific and Industrial Research Organisation (CSIRO) and the University of New England School of Environmental and Rural Science project expense support. The first author was supported by the Sheep Cooperative Research Centre (Sheep CRC) and the Commonwealth Government. The funders had no role in study design, data collection and analysis, decision to publish, or preparation of the manuscript.

==============================
Understanding the effects of repeated testing on behaviour is essential for behavioural tests that are re-applied to the same individuals for research and welfare assessment purposes. Assessing the repeatability of behaviour can also help us understand the influence of persistent traits vs transient states on animal responses during testing. This study examined the repeatability of behavioural responses in an attention bias test developed for sheep as a measure of affective state. Sheep were assessed in the attention bias test three times (n = 81 sheep), with testing occurring at intervals of 1 year then 2 weeks. During testing, individual sheep were exposed to a dog located behind a window for 3 s in a 4 × 4 m arena, then the dog was obscured from view, removed and sheep behaviours were recorded for 180 s. We hypothesised that behaviours in the test would have moderate-high repeatability but that the mean behavioural responses would change over consecutive trials as sheep habituated to the test environment. To estimate repeatability, data were modelled using restricted maximum likelihood linear mixed-effects models, fitting animal ID as a random effect. Vigilance behaviour, defined as having the head at or above shoulder height, was moderately repeatable (r = 0.58). Latency to eat (r = 0.20) and duration spent looking towards the previous location of the dog (attention to the dog wall) (r = 0.08) had low repeatability. Mean latency to eat did not differ significantly between trials (P = 0.2) and mean vigilance behaviour tended to decrease over the trials (P = 0.07). Mean duration of attention to the dog wall significantly decreased across the trials (P < 0.001), while mean zones crossed increased (P < 0.001), as did behaviours directed towards the exit door such as duration in proximity and pawing at the door. Overall, vigilance behaviour was moderately repeatable, suggesting it may have been driven by temperament or personality traits, while attention and feeding behaviours may have been more influenced by transient affective states or other factors, however further research is needed to better tease apart these potential effects. Sheep demonstrated some habituation to the test over consecutive trials. Care should therefore be taken during future application of the test to ensure all animals undergoing attention bias testing have equivalent experience for a valid interpretation of their relative behavioural responses.

Introduction

With growing recognition of animal sentience and emotions, the emotional or affective states of animals are increasingly being considered as a vital component of animal welfare. There are several frameworks for conceptualising and studying affective states in non-human animals; however, for the purpose of this study we will consider the framework described by Mendl, Burman & Paul (2010). Affect can be conceptualised as a location within a two-dimensional space delineated by the dimensions of valence and arousal; where valence describes the positivity or negativity of a state and arousal describes the intensity or level of activation (Mendl, Burman & Paul, 2010). Affective states include short-term emotions, which are triggered by specific events, as well as longer-term moods, which can be thought of as the running mean of an animal’s position within the affective space (Mendl, Burman & Paul, 2010; Kremer et al., 2020). Trait affect can be considered an aspect of animal personality or temperament that describes an animal’s propensity to experience a particular affective state (Boissy & Erhard, 2014). The concepts of personality and temperament broadly refer to the consistency of an animal’s behavioural responses across time and/or situations or contexts (Réale et al., 2007). Together, the personality of an animal and its transient emotions and moods combine to determine the way in which the animal responds to environmental stimuli (Finkemeier, Langbein & Puppe, 2018).

One method that has shown promise for determining affective states in animals is the assessment of affect-driven attention biases (Crump, Arnott & Bethell, 2018). An affect-driven attention bias is where an individual alters their allocation of attention towards certain types of information depending on their affective states (Bradley et al., 1995; Bradley, Mogg & Lee, 1997; Bar-Haim et al., 2007). Lee et al. (2016) developed a method to assess attention bias to a perceived threat in sheep, by measuring the variability between animals in allocation of attention towards a predator. Various versions of this method were shown to be influenced by short-term pharmacological manipulations that induced anxiety-like (Lee et al., 2016; Monk et al., 2018b) and depression-like affective states (Monk et al., 2018a). These studies suggest the method can provide information on the affective state of the sheep, but questions remain as to which aspects of affective state the test may be able to assess.

Studies in humans have associated attention bias to threat with differences in both trait (Bar-Haim et al., 2007) and state anxiety (Quigley et al., 2012; Nelson et al., 2015). However, the relationship between personality or temperament traits and responses during attention bias testing has been scarcely examined in animals (parrots: Cussen & Mench, 2014; pigs: Luo et al., 2019; macaques: Howarth et al., 2021; cows: Kremer et al., 2021) and not yet studied in sheep. Studies examining repeatability of animal responses during consecutive attention bias tests can begin to provide information on which aspects of an animal’s affective state or personality most strongly influence animal behaviour in the test, by supporting or opposing temporal stability of behaviour, as a key aspect of personality. This information will allow for a clearer interpretation of animal responses during future use of the test.

Understanding the potential effects of habituation, extinction and sensitization during repeated exposure to a test is also important when assessing the same individuals before and after an experimental manipulation or when using a test to track their well-being over time (Erhard, Elston & Davidson, 2006; Doyle et al., 2010). This is particularly important for tests that rely on novelty as part of the procedure and in tests carried out in extinction, where a negative stimulus is not followed by a negative consequence (Erhard, Elston & Davidson, 2006). For a test that is designed to compare relative responses within a given population of animals, a change in mean reactivity over consecutive tests may be acceptable if the tested population have equivalent prior experience with the test. However, if the rates at which animals habituate or become sensitized to a particular test situation differ, then it may not be appropriate to use the test for repeated assessments of individuals.

The current study aimed to examine the effect of repeated testing on sheep responses in an attention bias test. Vigilance behaviour has been previously considered as a measure of trait fearfulness (Beauchamp, 2017; Monk et al., 2018a) and locomotive behaviours are shown to be repeatable within similar contexts such as the arena test (Kilgour & Szantar-Coddington, 1995; Kilgour, 1998; Wolf et al., 2008). Consistency has also been shown in feed-directed behaviour within a similar context for cows (Kremer et al., 2021). Thus, we hypothesised that behaviours in the attention bias test would have moderate-high repeatability. We hypothesised that the mean behavioural responses during testing would change between trials, as animals became habituated to the novel test arena and learnt that the potential threat was not further reinforced (Erhard, Elston & Davidson, 2006; Doyle et al., 2010). To test these hypotheses we analysed a subset of data collected as part of a broader experiment, which assessed attention bias in adult ewes on three occasions, using the test method described by Monk et al. (2018b).

Materials and Methods

Animal ethics

The protocol and conduct of the study were approved by the University of New England Animal Ethics Committee, under the New South Wales Animal Research Act 1985 (Animal Research Authority numbers 16-003 and 17-015).

Experimental design

This experiment was conducted over 2 years from March 2016 to April 2017. The current study reports on a subset of data that were generated as part of a larger experiment, which aimed to examine the relationships between a suite of behavioural tests and physiological measures in sheep. At the beginning of the experiment, 100 sheep individually underwent the following consecutive procedures, in the listed order, over a period of approximately 14 min: blood sampling, attention bias testing, arena testing, isolation box testing, eye temperature measurement, flight speed assessment and a second blood sampling. The details for each procedure are briefly given below. Sheep were then returned to the farm and were managed under typical Australian extensive farming conditions. Management included a low frequency of periodic mustering using dogs for monitoring health and to carry out husbandry procedures (drenching, crutching, shearing, etc.) together with regular monitoring of animals while they grazed freely at pasture. After 1 year, the circuit of testing procedures was repeated on 81 of the sheep that were available to be retested. Two weeks after completing the second testing circuit, all sheep were tested in the attention bias test a third time. No additional procedures occurred at the time of the third attention bias test repetition. The current article focuses only on the attention bias tests and does not report on any of the additional behavioural tests and measures.

Animal details

One hundred adult ewes were used in this study. The sheep belonged to the Sheep CRC Information Nucleus Flock in Armidale, NSW, Australia. The flock included Merino sheep (n = 73) and Merino × Border Leicester (or similar) sheep (n = 27). The animals used in this study were selected from a larger population of 340 ewes, based on their immune competence phenotype, as assessed by Hine et al. (2017). The population was ranked on immune competence, then the top 50 (High) and bottom 50 (Low) ranked ewes were selected for the experiment. Sheep were approximately 4 years old and weighed an average of 52.5 ± 6.4 kg at the beginning of the experiment. The average weight was 57.1 ± 6.6 kg at the second time of testing 1 year later. All sheep were raised together under extensive farming conditions and were housed at pasture for the duration of the study. The sheep had regular contact with humans and dogs throughout their lives and had previous exposure to behavioural tests such as the isolation box test, but no prior experience with attention bias testing.

Attention bias test

The current study used the same attention bias test arena and testing procedure as described by Monk et al. (2018b). The test comprised a 4 × 4.2 m arena with 1.8 m high opaque walls (Fig. 1). Approximately 1.5 kg of lucerne hay was positioned in the middle of the arena. A small window was positioned on one side of the arena, which could be completely obscured by a retractable opaque cover. A stationary kelpie-x border collie dog stood behind the open window at the beginning of the test as the sheep entered the arena. The dog was held on a leash by a human who may have been partially visible to the sheep. The dog was visible for 3 s, then the window was covered and the dog was removed. A timer began once the window was fully covered and sheep remained in the test for a further 180 s while behaviours were recorded. The behavioural responses recorded during testing are summarised in Table 1. Behaviours were captured by a video camera positioned approximately 4 m above the ground next to the arena (Fig. 1). The camera was connected to a digital video recorder and captured by IVMS4200 software from Hangzhou Hikvision Digital Technology Co., Ltd. (Hangzhou, China). The video footage was viewed in real time by an observer positioned approximately 5 m away from the test arena to monitor animal behaviour during testing and start the timer. Behaviours were collated from video footage by two observers using The Observer XT 12.0 (Noldus Information Technology, Wageningen, The Netherlands). The observations were divided between the two observers. Inter-observer reliability was calculated using Microsoft Excel on 22 animals that were scored by both observers. Across all behaviours, the mean Pearson correlation between the two observers was 0.99, with a range of 0.97 to 1.0 (P < 0.001), while the mean difference in scores given by each observer was 0.4, with a range of 0 to 0.9 across the behaviours. A 3 × 4 grid was overlaid onto the video footage for calculation of zones crossed and time spent in the zone closest to the exit (Fig. 1). Prior to testing, sheep were held in a paddock with limited feed available overnight but were given ad lib access to water.

Figure 1 Schematic diagram of the attention bias test.

The symbol “*” denotes the position of a camera. The walls of the attention bias test were 1.8 m high and covered in opaque matting. The dashed lines denote the zones overlaid onto video footage, these lines were not physically marked during testing. The shaded zone denotes the zone for which duration standing near door was characterised.

Table 1 Ethogram of behaviours recorded during the attention bias test (Monk et al., 2018b).

Behaviour	Definition	
Vigilance	Duration with the head at or above shoulder height.	
Attention to dog wall	Duration looking towards the closed dog window with binocular vision, determined based on head orientation, for the first 60 s of testing.	
Eating	An eating bout began when a sheep took a bite of hay and ended when the sheep became vigilant or moved approximately 30 cm away from the hay. Number of eating bouts (eating frequency) and latency to eat from the time the dog window closed were recorded. Latency to eat was recorded at 180 s if a sheep did not eat during the test.	
Zones crossed	Number of zones crossed with both front feet in a new zone, or with one front foot in the zone and the other on the line.	
Standing near door	Duration standing in the upper corner zone closest to the exit.	
Pawing	Lifting a front foot and making contact with the door in a pawing motion or digging at the ground in front of the door.	

Additional tests and measures

The following circuit of consecutive tests and measures were conducted twice, at an interval of 1 year, which included the first two repetitions of the attention bias test reported in this article. The testing circuits aimed to assess the relationships between different aspects of sheep behaviour and physiological stress responses. Prior to undergoing the testing circuits each year, all sheep were weighed and had numbers painted on their rumps for individual identification. Internal body temperature was recorded throughout testing using Thermochron iButtons® (Embedded Data Systems, Lawrenceburg, KY, USA), which were attached to blank, progesterone-free Controlled Internal Drug Release devices (CIDR®; Zoetis, Melbourne, Australia) as described by Lea et al. (2008). An iButton was inserted into the vagina of each sheep 1 day prior to circuit testing and was removed after the testing circuit had been completed. No iButtons were used during the third attention bias repetition.

At the beginning of the 14 min testing circuit, baseline blood samples were collected via jugular venepuncture by an experienced handler. Sheep then underwent attention bias testing. Immediately after attention bias testing, sheep were moved into an arena test similar to that described by Murphy et al. (1994) for 3 min, to measure approach/avoidance conflict as an indicator of fear of humans. The 12 m × 6 m arena contained a small pen of three conspecifics, in front of which a stationary human sat quietly on a stool. After arena testing, sheep underwent isolation box testing for 30 s (Murphy et al., 1994; Bickell et al., 2009). Sheep were then restrained in a handling crate for 5 min and their eye temperature recorded using an infra-red thermography camera (ThermaCam T640; FLIR Systems AB, Danderyd, Sweden). Sheep were then moved into a weigh crate for assessment of flight speed over a distance of approximately 2 m using infrared sensors (Ruddweigh Australia Pty Ltd., Guyra, Australia). Blood samples were taken again at the end of the circuit, immediately after flight speed assessment.

Statistical analysis

Data were analysed using R version 3.5.1 (R Core Team, 2018). In order to determine repeatability estimates, data were first modelled with restricted maximum likelihood (REML) mixed-effects models using the package lme4 (Bates et al., 2015). Eating frequency data were modelled fitting a Poisson distribution, all other models were fitted with a Gaussian distribution. All model residuals were checked using visual assessment of residuals vs fitted values plots and histograms. A square root transformation was applied to time standing-near-door data. Latency to eat data could not be transformed to meet normality assumptions due to the censoring of data at 180 s. Therefore, repeatability estimates were obtained from the model fitting a Gaussian distribution to the raw data, but comparison of mean responses across trials were made using a Cox proportional hazards model, as described below. Sheep ID was fitted as a random effect in all models to account for repeated measures, following the procedure described by Field, Miles & Field (2012). Breed (Merino or Maternal) and immune grouping (High or Low) were fitted as fixed effects in all models and weight at the beginning of the experiment was fitted as a covariate. Immune grouping and weight were subsequently removed from all models using a backward stepwise reduction, considering the lowest Akaike Information Criterion and Bayesian Information Criterion. Breed was retained for latency to eat and eating frequency data but was removed from all other models.

Repeatability (r) was then calculated from the between-animal ( σB2) and within-animal ( σW2) components of variance for each behaviour as r=σB2/(σB2+σW2) (Bell, Hankison & Laskowski, 2009; Dingemanse et al., 2010). Repeatability estimates were confirmed and uncertainty in the estimates was quantified using the rptR package (Stoffel, Nakagawa & Schielzeth, 2017). This package relies on the mixed-effects models fitted using the lme4 package and uses parametric bootstrapping for estimation of confidence intervals and standard errors. Uncertainty estimates were based on 1,000 bootstrapping runs. Repeatability estimates were also calculated in the same way for subsets of the dataset, that examined data from trials 1 and 2 only, then trials 2 and 3 only. To better examine the relative performance of individuals over repetitions, repeatability estimates across all trials were also obtained using ranked data instead of raw data, using the procedure described above. Tied values were averaged when determining rankings. Repeatability estimates of <0.4 were considered to be low, 0.4–0.7 to be moderate and >0.7 to be high, as suggested by Harper (1994). Repeatability estimates were made for all behaviours except for pawing, due to a low occurrence of this behaviour.

To compare mean behavioural responses across the trials, all data except latency to eat were modelled as described above, using maximum likelihood mixed-effects models instead of REML, and fitting trial number as an additional fixed effect. Post hoc multiple comparisons were conducted using a Tukey method for adjustment of P-values. P values < 0.05 were considered significant while 0.05 < P < 0.1 were considered as tendencies. Latency to eat data were analysed with Cox’s proportional hazards model using the survival package (Therneau & Grambsch, 2000; Therneau, 2015). Animals that failed to eat within 180 s were deemed as censored results. The number of animals that pawed at the exit door were analysed using a Fisher’s Exact Test, post hoc multiple comparisons were performed using the package rcompanion (Mangiafico, 2018).

Results

Raw behavioural data are summarised in Fig. 2. Vigilance was moderately repeatable and the most repeatable behaviour that was analysed, while attention to the dog wall was least repeatable across all trials (Table 2). Attention to the dog wall repeatability estimates increased when using ranked data, compared to unranked data, although the estimates remained low regardless (Table 2). Vigilance, zones crossed and attention to the dog wall data were more repeatable between trials 2 and 3 than between trials 1 and 2 (Table 2). The feeding behaviours and time standing near the door were more repeatable between trials 1 and 2 than between trials 2 and 3 (Table 2). Mean duration of time spent displaying vigilance and latency to eat did not change significantly across all trials (Table 3). Duration of attention to the dog wall decreased over the trials, while zones crossed, time standing near the door and pawing at the door increased over the trials (Table 3).

Figure 2 Boxplots displaying the distribution of observed behavioural data during attention bias testing over three repeated tests.

Boxplots display the median values, the interquartile range (IQR) and range of data within 1.5× the IQR. The dots represent raw data for individual sheep within each trial. We note that the plot axes are scaled differently to more clearly display the data within each observed variable.

Table 2 Repeatability estimates for behavioural responses in the attention bias tests, across all trials and between pairs of adjacent trials using raw data and across all trials using ranked data.

Repeatability was estimated using the package rptR. Breed was retained as a fixed effect in the models for eating frequency and latency to eat only, based on AIC and BIC.

Behaviour	All trials	Trials 1 and 2 only	Trials 2 and 3 only	All trials (ranked data)	
r	s.e.	CI	r	s.e.	CI	r	s.e.	CI	r	s.e.	CI	
Vigilance (s)	0.58	0.06	[0.44–0.68]	0.55	0.08	[0.38–0.68]	0.70	0.06	[0.57–0.79]	0.56	0.06	[0.43–0.67]	
Zones crossed (n)	0.47	0.07	[0.34–0.59]	0.43	0.09	[0.23–0.59]	0.59	0.07	[0.44–0.72]	0.46	0.07	[0.32–0.59]	
Eating frequency (n)1	0.36	0.12	[0.11–0.56]	0.50	0.17	[0.13–0.76]	0.21	0.13	[0.00–0.48]	0.35	0.07	[0.19–0.47]	
Time at door (s)	0.29	0.07	[0.14–0.43]	0.46	0.09	[0.27–0.61]	0.35	0.09	[0.15–0.52]	0.36	0.07	[0.21–0.49]	
Latency to eat (s)	0.20	0.07	[0.07–0.34]	0.31	0.10	[0.10–0.50]	0.12	0.10	[0.00–0.34]	0.23	0.07	[0.09–0.37]	
Attention to dog wall (s)	0.08	0.06	[0.00–0.22]	0.05	0.08	[0.00–0.27]	0.28	0.10	[0.10–0.46]	0.33	0.07	[0.17–0.46]	
Note:

1 Original scale approximations for repeatability estimates are presented.

Table 3 Mean ± s.e.m. data for behavioural responses in the attention bias tests across three trials.

Behaviour	Mean ± s.e.m.	Analysis	Test value (df)	P	
Trial 1	Trial 2	Trial 3	
Vigilance (s)	157.2 ± 1.8	154 ± 2.5	152.5 ± 2.4	lmer	X2 (2) = 5.27	0.072	
Zones crossed (n)	21.5 ± 1.3a	26.1 ± 2b	30 ± 2b	lmer	X2 (2) = 21.3	<0.001	
Eating frequency (n)1	0.7 ± 0.1 (29)a	1.3 ± 0.2 (37)b	0.8 ± 0.2 (27)a	glmer	X2 (2) = 15.6	<0.001	
Time at door (s)	33.4 ± 3.9a	42.7 ± 4.1a	67 ± 4.7b	lmer	X2 (2) = 54.6	<0.001	
Latency to eat (s)	139 ± 6.7	121.3 ± 8	140.7 ± 7.1	Cox	LR (2) = 3.49	0.2	
Attention to dog wall (s)	33.8 ± 0.8a	26.7 ± 1.1b	19.5 ± 1.0c	lmer	X2 (2) = 148	<0.001	
Pawing at door (n)1	0.04 ± 0.0 (3)a	0.14 ± 0.0 (11)ab	0.16 ± 0.0 (13)b	FET	N/A	0.02	
Notes:

1 Raw numbers of animals that exhibited the behaviour are given in parentheses.

a,b,cDifferent superscripts within rows indicate significant differences between trials as determined using post hoc analyses. lmer, linear mixed-effects model; glmer, generalised linear mixed effects model with Poisson distribution; Cox, Cox proportional hazards model; FET, Fisher’s exact test; LR, likelihood ratio.

Discussion

Repeatability estimates for vigilance behaviour were moderate and mean vigilance duration did not change significantly over the three trials, which partially supports our hypotheses. The observed consistency of vigilance behaviour over time supports suggestions that vigilance may be used as a measure of a fearfulness trait in sheep and other grazing ruminants across a range of contexts (Beauchamp, 2017), as an important and innate behavioural response to both isolation and the threat of predation (Frid, 1997; Kendrick, 2008). These findings are also supported by Kremer et al. (2021), who found vigilance behaviour was related to fearful and social personality traits in dairy cows within a similar attention bias test arena, and by Welp et al. (2004), who found moderate correlations between vigilance responses of individual cows across different threat contexts. Nevertheless, previous attention bias studies have demonstrated that mean vigilance duration can be altered by pharmacological interventions that temporarily alter affective states (Lee et al., 2016; Monk et al., 2018b, 2018a). It is therefore important to consider that both affective states and trait affect may have a role in determining the expression of vigilance behaviour, although our findings support its use as a measure of trait anxiety or fearfulness in a context where no other affect-altering treatments have been applied.

Duration of attention to the dog wall had low repeatability, which contrasted with our hypotheses and other studies in humans and animals. In humans, studies have suggested attention biases measured as “looking” and “gaze” can indicate trait affect (Bar-Haim et al., 2007). Using preferential-looking time tasks in macaques, Howarth et al. (2021) found stable individual differences in baseline attention, characterized by time looking at a threat face and a neutral-threat face pair. Responses also remained consistent in a small subset of the macaques that were assessed again after several years (n = 18), supporting the use of their attention bias test method as a measure of trait affect. In dairy heifers, using an attention bias test similar to that of the current study, Kremer et al. (2021) found some relationships between fearful and social personality traits and attention to threat. However, they found no significant relation between threat-directed behaviours within individuals across two attention bias tests, suggesting threat-directed behaviours such as looking duration may not measure trait affect using this test paradigm in livestock. This may relate to difficulties in characterising attention through head orientation alone in species with a wide field of vision (Piggins & Phillips, 1996). Overall, our findings suggest that looking behaviour, as we have defined it, may not be a reliable indicator of personality or trait affect in this testing context.

Food related behaviours showed both poor repeatability and inconsistency of mean responses across the trials, with eating frequency peaking in trial 2. The repeatability of feeding behaviours was considerably lower between trials 2 and 3 than trials 1 and 2, which was unexpected considering the shorter time interval between the later trials. In dairy heifers, Kremer et al. (2021) found feeding-directed behaviours in an attention bias test were positively correlated across two exposures to the test irrespective of experimental conditions, contrasting with our findings. Notably, Kremer et al. (2021) used a different threatening stimulus and positioned the food in a different location compared to the current study, which may have impacted feeding behaviour. Further, a number of other factors are known to influence feeding behaviour which may have confounded our results. While feeding schedules were consistent within the trial periods, we did not assess pasture availability, pasture quality or ewe body condition leading into the trial or between years 1 and 2. These factors may have been important as chronic food restriction has been shown to increase feeding motivation, perceived hunger and attention bias towards food in sheep (Stockman et al., 2013; Verbeek, Ferguson & Lee, 2014). Sheep are also known to change their patterns of eating activity based on novelty of feed in order to seek out a more diverse diet (Favreau, Ginane & Baumont, 2010). During the circuit testing periods, sheep were fed a small amount of highly palatable lucerne hay to supplement the smaller pastures in which they were housed. Thus, familiarity with the feed may have contributed to a reduction in eating frequency during the third attention bias test. Overall, the lack of consistency in feeding behaviour across tests demonstrates a shortfall in using feed as a positive stimulus for attention bias testing, due to the potentially confounding effect of hunger and feeding motivation on behavioural responses during testing.

It should be considered that the first two repetitions of the attention bias test occurred within a circuit of other behavioural tests and physiological measures, that may have impacted animal behaviour during the attention bias test. The attention bias test was the first behavioural test to be conducted within the circuit, reducing the potential impact of other tests and measures. However, it is possible that the memory of the first testing circuit impacted animal responses during the second and third repetitions, if sheep were anticipating the additional stress caused by subsequent tests. This may have resulted in an altered affective state during the later repetitions, which could explain the poor repeatability of attention and feeding behaviours but would further support suggestions that vigilance may provide a measure of a more consistent personality trait.

Mean duration of attention to the dog wall significantly decreased over consecutive trials, while zones crossed and time standing at the door increased. The negative stimulus of a predator threat was not followed by a negative consequence in this testing context, resulting in extinction of the unreinforced behaviour, as demonstrated in other studies (Erhard, Elston & Davidson, 2006; Doyle et al., 2010). A shift in attention towards the door and increased zones crossed may indicate a reallocation of attentional resources from the threat of predation towards escaping the arena to reunite with their flock-mates. In both cases, vigilance behaviour, as defined by having the head at or above shoulder height, may benefit the sheep, potentially explaining why the mean vigilance behaviour remained consistent across the trials, while the duration of threat-directed behaviour decreased. Given that an animal’s responses change with repeated attention bias testing, it will be important to ensure that all sheep being tested at a given time have had the same level of experience with the test, so that the influence of habituation or extinction does not confound interpretation of comparative animal responses.

There is evidence that sheep habituated to the test over consecutive trials as a cohort, but there was also variation in the rate at which individuals habituated to the test. The repeatability estimates obtained using ranked data were higher for attention to the dog wall and time standing near the door compared to the repeatability estimates obtained from raw data. This increased consistency of rankings compared to raw values suggests that sheep learnt the predator threat was not further reinforced at different rates. It might also be considered that the rate at which animals learn or acclimate to new environments may itself have welfare implications within livestock production systems (Wechsler & Lea, 2007; Monk et al., 2018c). Considering this interpretation, it may be valuable for additional studies to examine the rates of change in attention towards the dog wall over consecutive tests as an additional aspect of animal personality or learning ability.

The repeatability estimates for zones crossed were moderate and comparable to those found for measures of activity during other arena tests in sheep that measured conflict between a human and conspecifics (Murphy et al., 1994; Kilgour & Szantar-Coddington, 1995; Kilgour, 1998; Wolf et al., 2008). The observed consistency suggests that measures of activity may have been influenced by underlying temperament or personality traits, however further analyses comparing responses in the attention bias test across other testing contexts are needed to confirm this suggestion. This suggestion is, however, supported by other studies that have found consistency in activity not only over time, but also across testing contexts, including arenas that presented different types of stimuli (Beausoleil et al., 2012) and measures taken across an open field, novel object, runway and attention bias test (Kremer et al., 2021).

Measures of activity in sheep within the context of the arena test have also been shown to be heritable (Wolf et al., 2008). Given the moderate to high repeatability of vigilance behaviour, it may be possible that this behavioural response has a genetic basis as well, which can be incorporated into selection programs (Dohm, 2002). Selection for calmer and less fearful animals can have both welfare and production benefits, as calmer animals become easier to handle and interactions with humans cause them less stress. However, it should be considered that attention biases are very context specific and that responses to live dogs may not necessarily reflect responses towards other stimuli such as humans (Beausoleil, Stafford & Mellor, 2005). Further, increased vigilance towards predators may be beneficial in an extensive farming context where the threat of predation can have a significant impact on animal welfare and production outcomes. Further work is therefore required to better understand how vigilance and activity might relate to animal welfare or production outcomes more broadly before determining whether this would be a trait of interest for which to select.

Sheep in the study were drawn from a population that had been phenotyped for immune competence. This trait is associated with a range of health and productivity outcomes in sheep (Hine et al., 2022), beef (Hine et al., 2019, 2021) and dairy cattle (Mallard et al., 2015). In beef cattle, immune competence has moderate genetic but low phenotypic correlations with flight speed (Hine et al., 2019) which has in turn been associated with some behaviours in the attention bias test (Lee et al., 2018). Associations between indicators of affective state, immune function and health outcomes have been observed in a number of species (Walker et al., 2012; Düpjan & Dawkins, 2022). In the current study, there was no significant effect of high vs low immune competence when analysed as a categorical factor on behaviours in the attention bias test. Links between transient affective states and persistent personality traits, immune function, and health outcomes are complex and require further study (Düpjan & Dawkins, 2022).

This experimental design cannot definitively show any behaviour to be a measure of personality or trait affect, without applying conditions that alter affective state between repetitions of the attention bias test and examining the consistency of behaviour across contexts. The observed low repeatability of attention and feeding behaviours does, however, suggest these behaviours are not strongly indicative of personality. Teasing apart the potential effects of emotions, moods and environmental conditions on these behaviours will be important for further refinement of the attention bias test (Vögeli et al., 2015). If these behavioural responses are readily influenced by known or unknown internal and external factors (e.g., recent handling or housing, weather, noise, familiarity with dogs, hunger etc.), the measures may have little use for the assessment of animal welfare. If these measures capture transient, labile emotional states caused by short-term events occurring immediately prior to testing, the measures may have some limited applications, such as in research to determine the effect of specific events or environments on animal affect. If the test can measure longer term moods, this would provide information on the cumulative effect of recent events that have impacted on the animal. A measure of mood could be applied in research settings as well and would be ideal for application as an on-farm welfare assessment tool. Further studies are therefore needed to tease apart the potential effects of affect, personality and other external factors on behaviour during attention bias testing.

Conclusions

Some behaviours in the attention bias test were more repeatable than others. Vigilance and zones crossed behaviours were most repeatable and are likely to be more strongly driven by stable personality or temperament traits, although examination of the consistency of these behaviours across contexts is still needed. Attention and feeding behaviours were least repeatable and are likely to be more strongly influenced by emotions, moods or unidentified internal and external effects. It is possible the attention bias test can be used as a measure of both trait and state affect by considering different behaviours during testing, however, further work is needed to better tease apart the variable effects of discrete emotional states, moods and personality or temperament on animal responses during testing. This will be essential to determine how best to apply the test in future and for a clear interpretation of animal responses. Some of the mean behavioural responses of sheep changed between the trials. Thus, when applying the test in future, it is recommended that all animals have the same level of experience with the attention bias test to ensure the effects of habituation and extinction do not confound the comparison of individual responses.

Supplemental Information

Supplemental Information 1 Dataset collected on 81 Merino or Maternal breed ewes that underwent attention bias testing 3 times.

Raw data are provided in the first sheet and a key is provided in the second sheet to explain the variable names.

Click here for additional data file.

Thank you to the staff and students at CSIRO for their assistance during the experiments: Jim Lea, Sue Belson, Tim Dyall, Troy Kalinowski, Grant Uphill, Brian Dennison, Jody McNally, Jess Burton and Koli the dog. Thank you to Geoff Hinch and Brad Hine for providing supervision and advice for the broader project.

Additional Information and Declarations

Competing Interests

Author Contributions

Animal Ethics

Data Availability

The authors declare that they have no competing interests.

Jessica E. Monk conceived and designed the experiments, performed the experiments, analyzed the data, prepared figures and/or tables, authored or reviewed drafts of the article, and approved the final draft.

Ian G. Colditz conceived and designed the experiments, analyzed the data, authored or reviewed drafts of the article, and approved the final draft.

Sam Clark conceived and designed the experiments, analyzed the data, authored or reviewed drafts of the article, and approved the final draft.

Caroline Lee conceived and designed the experiments, authored or reviewed drafts of the article, and approved the final draft.

The following information was supplied relating to ethical approvals (i.e., approving body and any reference numbers):

The University of New England Animal Ethics Committee provided full approval for this research (Animal Research Authority numbers 16-003 and 17-015).

The following information was supplied regarding data availability:

The raw data are available in the Supplemental File.

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
