# Peer review of "Repeatability of an attention bias test for sheep suggests variable influence of state and trait affect on behaviour"

_PeerJ, doi:10.7717/peerj.14730_

## Round 0.1 · original submission · Minor Revisions

Three expert reviewers have each praised the readability and value of your study. Each has recommended minor revisions, hence my minor revision decision. However, each has identified unique areas in need of improvement and clarity. The reviewers have been very thoughtful and thorough and I would encourage you to provide all of the requested clarifications and consider their suggestions carefully in your revision.

I agree with the reviewers that the general approach is of value, particularly attempting to tease apart persistent versus transient states on responses during an attention bias test. I read this as a plan to correlate different measures of mood state with performance in an attention bias test. However, even though this goal appears to be what is stated in the abstract, your statistical approach does not provide any test of this question. You report tests of the repeatability of each measure, but you do not use these other measures to predict attention bias results, so I am a bit confused about the aim of the study given this. Although I agree with the reviewers that the paper is well-written, I think the exact goals could be framed more clearly.

Could you please clarify if any of the current study data were already reported in the previous study (Monk et al., 2018) or if the additional data will be reported in a future publication? It is important that we are very transparent about any overlap in published data.

Line 224 “which” should be “that.”

I do not think you need to apply a PCA approach to your data.

Thank you for submitting your interesting work to PeerJ.

Reviewer 1 ·

Basic reporting

The manuscript is well written, organised, and easy to follow. It was a pleasure to read it and I think that it provides insightful methodological advice to be taken when an attentional bias paradigm is used to make a solid interpretation of the behavioural responses observed during the test. I noticed that quite often the words “state” and “trait” are not defined such as emotional state or personality trait in the manuscript. Can this cause some confusion?
In addition, I was wondering whether a different statistical approach would be better. My concern stems from the fact that the six behaviours measured could be summarised in fewer factors using a Principal Components Analysis on which later perform some kind of inferential statistics. I imagine that for, example, Vigilance behaviour and Attention to dog wall are highly correlated therefore likely to measure that same latent construct. This can be a useful approach to investigate if these behaviours are similar when we approach the study of personality in humans (such as the Big Five or Eysenck three factors traits). If I correctly remember Kremer at al., 2021 used a PCA statistical approach in a similar experiment in dairy heifer. Are there any reasons why the authors did not consider using a PCA analysis?

I have few minor points listed below.

At the beginning of the abstract, a sentence providing a frame to contextualise the topic of the manuscript or alternatively the importance of repeatability of behavioural responses in the ABT (attentional bias test) at general level (Why we should care) seems missing. Also, I am not sure if PeerJ allows to add statistical results in the abstract. Please check the guidelines. In my prospective this is a minor point, however, in the results section of the ms the authors did not report any statistics, which of course enhance the readability but because of this I think they can use a similar approach for the abstract.

Line 46-47 Not sure if this statement is factual. If we consider the initial paper when the cognitive bias was introduced in the animal welfare field (Harding et al 2004) we have already almost 20 years of research with several review papers (if I remember correctly 5) and two meta-analysis papers published; one of which focuses on the pharmacological effects and over 200 papers published so far. Obviously, in my opinion, we are at the stage where we are refining the methods developed more than navigating in complete darkness.

Line 64 Why using the word “discrete” here – it seems in opposition to the dimensional view of emotions described in the previous sentence? Plus, worth to cite more recent papers here, which describe exactly what the authors are explaining (Kremer et al., 2020…figure 3).

Line 133 I do not understand to what 81 refers to here.

Line 228 I do not know if the current info provided in the results section are sufficient and comply with the PeerJ guidelines.

Line 270 I would remove the word “discrete”

Experimental design

No comment

Validity of the findings

No comment

·

Basic reporting

No comment

Experimental design

No comment

Validity of the findings

No comment

Additional comments

Thank you to the authors for a well written manuscript with clearly outlined methods – greatly appreciated! This paper is a novel contribution to understanding and refining attention bias testing methods in livestock – and has an interesting discussion regarding interpretation of the behaviours as reflecting personality traits or transient / acute emotional states.

Abstract:
L19 – would be helpful to indicate over what period of time here
L23 – suggest you change ‘under the influence of’ to driven by personality
L37 – use ‘may have been’ as you did for suggesting may have been driven by personality.
Introduction:
L59 section – I find it a bit out of order to talk about the concept / framework of emotional states after you have mentioned methods how to measure it (and after this section you return to talking about attention bias). Suggest you reverse the order – talk about the framework, and then mention how to measure it, including your previous attention bias testing, and the more detail about attention bias.
L77 – indicate which animals these references are in
L100 – rankings of zones.. do you mean number of zones crossed?
Methods:
L134 – so the second round of attention bias testing occurred after 1 year? Just clarify exactly the duration between each test.
L147 –at the time of second testing
L142 – considered breed in stats?
L149 – how do you think this regular contact with dogs affected their responses in the test. Do you think you’d see the same results with a population that was unfamiliar with dogs? (to mention in discussion)
L158 – was a person behind the window with the dog? Did the dog remain motionless while window was open?
L160 – was there a person somewhere else that could see the test and thus start the timer?
L163 – not clear how many cameras you had or where they were positioned. I’m not sure how you were able to accurately determine the sheep was looking with both eyes at the dog. Also need to indicate number of observers scoring the behaviours from video, and reliability of observers.
L165 – not clear how you calculated latency to eat – put in Table 1. Was this from the moment the sheep entered the arena to when she began eating? Also indicate scoring if the sheep never went to the food.
Table 1 ethogram –
Attention to dog wall: is this scored based on the eyes or the head direction. Was this assessed from a video camera that focused on the sheep’s face? Why was this only for the first 60 s of the test. It wasn’t mentioned in the text.
Eating: how did you measure 30 cm from the hay.
Standing near door: from Figure 1 it seems to me there are 2 zones nearest the door. This also applies to L165 where you say zone closest to door. Is this the zone in the upper corner then?
Missing latency to eat.
L169 – this series of tests was done twice correct? Helpful to remind readers of this here, and that it was not done in connection with the third attention bias test.
L180 – at first read I assumed arena test meant novel arena test, but next sentence suggest this is a novel human / human approach test? It would be helpful to quickly state the purpose of this series of tests (novel human, isolation box, restraint, flight speed). I know they are not specific to this paper, but will help the reader to understand how all of these tests fit together and possible impacts they may have had on the animal’s responses in the attention bias test. Perhaps at the end of this section (L189) you could make a statement about how you expect these series of tests to affect the sheep’s responses to attention bias, if at all, in same year and the following year
Statistics:
L196 – all other variables were normally distributed? I assume this is why you transformed this data, to meet normality assumptions
L197 – can you indicate what your outcome variables are. It is not clear what you are testing with these models… is it the effect of immune status and breed on vigilance, attention to dog etc. ? Also it is not clear how you handled the repeated tests in these models?
L201 – using lowest AIC / BIC ?
L216 – here can you indicate which behaviours you calculated repeatability for.
L219 – okay so here you are considering the 3 repeated trials, so what did you do in the first set of models with the 3 trials.
L221 – it looks like from Table 2 that you only did the post doc comparisons on adjacent tests for repeatability but then for the means across trials you compare all comparisons? Why the same logic does not apply here?
L224 – what did you do with these censored results… included or excluded? Does this mean you scored these animals as having a latency of 180 s to eat? Here you refer to the test time as 180 seconds, but previously was 3 min
L226 – at end of stats, state declaration of P values.. it does not look like you consider tendencies.
Results:
Figure 2 – need an x axis label indicating test repeat.
L229-230 – can you indicate if these values are high, moderate, low repeatable.
L230 – attention to dog was least repeatable, do you think this is because it is harder to measure and thus more human measurement error?
L236 – `what is the difference between describing ‘consistency’ of vigilance and latency to eat, and describing ‘repeatability’ in the previous sentences? They are two different statistical approaches, which is fine, but I think there should be clarity about their meanings and why you do it.
Table 3 – Superscript 1 indicating number of animals… clarify that the analysis was done on the number of animals exhibiting the behaviour, not on the mean frequency per animal that you report in the table.
Figure 1 and 2 – not sure where the captions are?
Discussion:
L245 – can you say fearfulness trait instead of trait fearfulness (unless you are implying a different meaning than I am interpreting).
L249 – Kremer et al. 2021 study used adult cows not calves.
L254 – I think clarify when you say state you refer to acute affective state (correct?) and when you refer to trait you mean personality trait. You use these terms throughout so I think it would be good to clarify the meaning of these terms.
L258 – duration of looking toward dog… this is the same behaviour as attention to dog wall as in your ethogram? Just use the same terminology throughout.
L260 – trait affect refers to a long term affective state that could be a personality trait? I am not sure naïve readers will understand what the difference is between trait and state. (as you refer to it in L254). Also the previous sentence indicates low repeatability of attention to the dog, which would suggest it is not a stable behavioural response (i.e. trait). Perhaps add a clarification what low repeatability indicates
L269 – “insensitive to trait” – something missing here?
L270 – here you separate mood state from personality trait. So what is a ‘trait affect’ as you referred to it earlier in this paragraph. I understood you were suggesting long term mood states could be reflective of personality trait.
L271 – and so is the vigilance behaviour that was repeatable therefore more reflective of a personality trait? (of fearfulness?)
For the vigilance and looking at dog behaviours, how do you think regular contact with dogs affected the sheep responses in the test. Do you think you’d see the same results with a population that was unfamiliar with dogs?
L294-306 – I feel this paragraph is out of place, since the next paragraph returns to discussing a specific behavioural result. I think it would be better placed at the end of your discussion
L332 – what type of arena tests are these (since earlier you described an arena test that had a human in it). Similarly, on L336, what are these testing contexts?
L345 – this had me thinking about how responses would differ depending on the presentation of the dog and level of threat. Yours was a real dog, while Kremer and others have used a dog statue / model. You did not have the dog barking, while Kremer used an auditory cue of a growling dog. I wonder how these additional cues, and whether dog is real or fake, affect the responses. Can you comment on this.
L273-292 – I noticed that the position of your food was central, directly in front of the dog, with Kremer’s study had the food positioned more in a corner, and did not seem to be directly in front of the dog. Do you think food positioning would influence results (since you found no consistency and Kremer did?). Also, would it matter if you started the test once the animals reached the food (so all animals had the experience of trying the food – then the threat appears – then measure latency to return to food).
Limitations of your study, and generally of using attention bias tests – how difficult is it to actually reliably measure these behaviours. Attention to the dog seems it would be more difficult to determine exactly where attention is directed – so is there human measurement error here? Would this contribute to lower repeatability? I am thinking of how and why vigilance behaviours would be high repeatable and attention to threat, which we assume is a related behaviour, is low repeatable.
I think also a comment on the design of attention bias tests is needed, as I mention above, that there are inconsistencies in how people do them – with a real dog, fake dog, size of dog, type of stimuli (visual, auditory, olfactory) – and when the threat is presented, the context with or without other animals.

·

Basic reporting

This manuscript is written in a clear, unambiguous way in excellent, professional English. The line of arguments is easy to follow, and supported by the relevant literature references. The manuscript is well structured, tables and figures are clear, and data made available. The chosen experimental design is suitable for testing the hypotheses (but see limitations stated below).

Experimental design

The manuscript presents original primary research well within the scope of this journal. The research question, namely whether or not the behaviour of sheep in an attention bias test is repeatable, is well defined and highly relevant given the ever increasing number of studies using cognitive bias tests as indicators of animal welfare. There certainly is a knowledge gap regarding what exactly bias tests measure: transient states of consistent traits. The methods section is detailed enough to allow for a replication of the study, and shows that the study was performed to a high scientific standard.

Validity of the findings

The data presented within the manuscript appear to be valid, as are the statistical analyses. However, I have one general issue with how the results are interpreted.
The authors claim to test whether behaviours measured in repeated attention bias tests reflect stable individual traits or transient states. However, they also investigate habituation to the testing, which would (and does, at least for some parameters investigated here) reduce repeatability. Does lack of habituation mean repeatability? Did the experimental design allow for a separation of the two? Temporal consistency is then argued to be indicative of the influence of personality or temperament. However, as the authors themselves acknowledge in the introduction, personality and temperament are defined as consistency over time AND contexts, and there was only one context tested in this study. Therefore, the authors should be very cautious in their interpretations, given that they cannot show consistency across contexts. Also, I see a risk of circular argumentation: all that is inconsistent is interpreted as influenced by short-term states, while all that is consistent is interpreted as influenced by personality or temperament. But the current study did not include any treatments or interventions that influenced short-term states, and as mentioned before, personality was not assessed and linked to the behaviours investigated herein, so we cannot know what influences the observed behaviours.
All this does not reduce the relevance of the manuscript, I simply would appreciate a more cautious interpretation of the results, clearly stating the limitations of the study.

Additional comments

Some more detailed comments:
ll. 24 ff: I did not get why you had different expectations regarding the repeatability of vigilance on the one hand and attention and latency to eat on the other, neither here nor later in the introduction (ll. 102 ff). Also, I would think that at least attention towards the previous location of the dog would also be an aspect of vigilance.
l. 70: Why not refer to the original article giving this definition instead of Finkemeier et al., who quote the definition proposed by Reale?
ll. 79 ff: Here you clearly state both the potential and limitations of your approach; please be as cautious in the discussion.
ll. 103 f: Why "studies" if there seems to be only one (Kremer et al.)? It's a pity that this hypothesis is not well based in the literature, as to me it is not logical that (not) feeding in a potentially threatening situation is not considered an aspect of vigilance.
ll. 105f: Again, why this expectation? Is this not vigilance, and if so why expect one aspect of vigilance to be a trait and another to be a state?
l. 108: How do you know? Why should the learning be different from other behaviours?
ll. 135 f: I understand that you used the opportunities provided by another study, but it seems a bit unfortunate that both the time betweween trials and the additional, potentially stressful procedures do vyry between trials. However, this is well discussed later.
ll. 144 ff: Again, the selection of the subjects seems a bit unfortunate, as it is a sample that is not representative but biased towards the extremes. But then again this is discussed later.
l. 201: Why both AIC and BIC? Also, in the legends only AIC is mentioned. Was BIC used where non-Gaussian distributions were assumed?
ll. 222 ff:Were theses tests also performed in R? Whicjh package/library?
l. 230: Be more consistent in your wording: you did not observe attention to the dog, but to the window where the dog had appeared. You use different wording (attention to dog, to wall, looking) throughout the manuscript.
ll. 269 ff: I do not think your data allow for such a conclusion, since you did not induce different discrete emotional states or moods, and hopefully made sure there were no temporary environmental effects that could have caused the observed inconsistency within subject.
ll. 289 ff: Then why do you still consider your design appropriate?
l. 297: In addition to external factors it yould also be internal factors (hunger, malnutrition, sickness, reproductive state, estrus stage,...).
ll. 305 f: I wholeheartedly agree; but then please be more careful in the abstract.
ll. 334 f: Personality is defined as consistency over time and contexts, and you only tested consistency over time. Therefore, unless you for example show that activity was consistent in the arena test, you cannot draw such a strong conclusion. Could you use the arena test data from trials 1 and 2 to support your case?
l. 368: dito
ll. 369 f: see above; other internal or external effects are possible, and none of them were tested here.
ll. 371 f: I tend to disagree. As mentioned before, you did not show personality/temperament as you did not test consistency over contexts, and arguing that inconsistencies show emotion/mood and consistencies show personality/temperament seems a circular argument as long as you do not compare animals with clearly different traits or states. In essence, please be more cautious with your conclusions and avoid circular argumentation.
ll. 374 f: I cannot thelp but think that this sounds like a call for validation of a tool the authors have previously used. Shouldn't the validation preceed the application? Of course, I know that this is what happens with most cognitive bias research, and I appreciate that the authors make an effort to contribute to the validation of their study design.

---

## Round 0.2 · accepted · Accept

Thank you for your thoughtful response to the previous round of reviews. I believe you have provided a balanced and nuanced interpretation of your results and the rationale and hypotheses for the study are now clear. I am happy to accept your paper for publication in PeerJ.